

# 1 Ensemble-based data assimilation improves hyperresolution 2 snowpack simulations in forests.

**Esteban Alonso-González[1], Adrian Harpold[2], Jessica D. Lundquist[3], Cara Piske[2,4],** 4 **Laura Sourp[5,6], Kristoffer Aalstad[7] and Simon Gascoin[5]**
[1]Instituto Pirenaico de Ecología, Consejo Superior de Investigaciones Científicas 6 (IPE- CSIC), Jaca, Spain.
[2]Department of Natural Resources and Environmental Science, University of 8 Nevada, Reno, Nevada, USA.
[3]Civil and Environmental Engineering, University of Washington, Seattle, 10 Washington, USA.
[4]Airborne Snow Observatories Inc. Mammoth Lakes, California, USA.
[5]Centre d'Etudes Spatiales de la Biosphère (CESBIO), CNES/CNRS/IRD/UT3 Paul 13 Sabatier, Toulouse, France.
[6]MAGELLIUM, Ramonville Saint-Agne, France
[7]Department of Geosciences, University of Oslo, Oslo, Norway.
Corresponding author: Esteban Alonso-González (alonsoe@ipe.csic.es)

## 17 Abstract

Snowpack dynamics play a key role in controlling hydrological and ecological 19 processes at various scales, but snow monitoring remains challenging. Data 20 assimilation techniques are emerging as promising tools to improve uncertain 21 snowpack simulations by fusing state-of-the-art numerical models with 22 information rich, but noisy observations. However, the occlusion of the ground 23 below the forest canopy limits the retrieval of snowpack information from remote 24 sensing tools. Remote sensing observations in these environments are spatially 25 incomplete, impeding the implementation of fully distributed data assimilation 26 techniques. Here we propose different experiments to propagate the information 27 obtained in forest clearings, where it is possible to retrieve observations, towards 28 the sub-canopy, where the point of view of remote sensors is occluded. The 29 experiments were conducted in forests within  Sagehen Creek watershed 30 (California, USA), by updating simulations conducted with the Flexible Snow Model 31 (FSM2) using airborne lidar snow data using the Multiple Snow data Assimilation



system (MuSA). The successful experiments improved the reference simulations significantly both in terms of validation metrics (correlation coefficient from R=0.1 to R ~0.8 on average) and spatial patterns. Data assimilation configurations using geographical distances and space of topographical dimensions, improved the reference run. However, those creating a space of synthetic coordinates by combining the spatiotemporal data assimilation with a principal components analysis did not show any improvement, even degrading some validation metrics. Future data assimilation initiatives would benefit from building specific localization functions that are able to model the spatial snowpack relationships at different resolutions.

## 1 Introduction

The seasonal snowpack is a crucial component in various ecological and hydrological processes in mountain areas and cold regions (Han et al., 2024; Slatyer et al., 2022)**,** covering over 47 million square kilometers of the northern hemisphere (Robinson & Frei, 2000) and 45% of global mountain areas (Gascoin et al., 2024),. It has significant implications for both the economy and ecology of these areas, as well as for downstream regions (Barnett et al., 2005; Qin et al., 2020; Sturm et al., 2017). However, accurately estimating the spatiotemporal dynamics of the snowpack, in particular the snow water equivalent (SWE), remains a challenging and unresolved issue (Tsang et al., 2022). These difficulties are only increased in forested terrain, due to the complex relationships between snowpack and canopy cover (Mazzotti, Essery, Moeser, et al., 2020).

The overlapping area between the snowpack and forested areas is estimated in at least 19% of the terrain in the northern hemisphere, only accounting for the boreal forest (Rutter et al., 2009). This estimation can only be higher considering the overlapping area in alpine forests. Snow beneath the canopy behaves differently than in open terrain (Dickerson-Lange et al., 2023; Safa et al., 2021; Varhola et al., 2010). One major process is the interception of snowfall in the forest canopy, limiting under canopy snow accumulation compared to clearings (Essery et al., 2003). The intercepted snow will either sublimate, drip as liquid water or unload as snow (Lundquist et al., 2021). In addition, the canopy cover changes the net radiation available to melt the snowpack, both by shading the snow surface and increasing the incoming longwave radiation (Lundquist et al., 2013). Generally, this leads to increased ablation under the canopy in warmer environments from longwave radiation compared to colder environments where shading from solar radiation causes less ablation under canopy (Lundquist et al., 2013). This relationship leads to differences between under canopy and open clearing



snowpack in most environments (Dickerson-Lange et al., 2017) that are
challenging to observe across complex terrain (Safa et al., 2021).
Direct observations of the snowpack under the forest are rare and challenging to
obtain (Kinar & Pomeroy, 2015). Deploying field based monitoring networks is a
complicated and expensive task. The harsh weather conditions of the remote areas
where the snowpack is present complicate the installation of monitoring networks,
with the number of automatic weather stations declining dramatically with higher
elevation (Matthews et al., 2020). Given the considerable complexity of the spatial
patterns of the seasonal snowpack, monitoring networks often suffer from a lack
of representativity (Herbert et al., 2024). In addition, monitoring SWE, which is the
key snow hydrological variable, is significantly more uncertain and costly to
measure than other variables such as snow depth, and remains an active research
topic (e.g., Gugerli et al., 2022; Orio-Alonso et al., 2023). The extensive spatial
extent of the seasonal snowpack and its temporal variability make monitoring
based on manual field campaigns challenging to deploy.
Remote sensing techniques are well established as snow cover monitoring tools
(Gascoin et al., 2024). Due to different remote sensing initiatives, it is possible to
monitor the dynamics of the snowpack even at continental scales at frequencies
approaching real time. Despite being traditionally restricted to the measurement
of snow cover properties such as the snow cover extent, fractional snow cover or
snow surface temperature, it is now possible to retrieve the snow depth, from
photogrammetry and lidar based sensors installed in traditional or remotely
piloted airborne platforms, or orbital sensors (Deschamps-Berger et al., 2020;
Harder et al., 2020; Painter et al., 2016). Unfortunately, most of these retrievals are
limited to observations in open terrain or clearings in forested areas, being limited
either spatially or temporally. Recent experiments based on airborne campaigns
have proven the potential of  X- and Ku-band SAR technology to retrieve SWE
(Montpetit et al., 2024; Singh et al., 2024), a technology that is expected to be
implemented in the next generation of satellites in the near future (Derksen et al.,
2021). Unfortunately obtaining SWE observations in dense forest areas will remain
problematic (Tsang et al., 2022). One partial solution to observing snow under the
canopy is with airborne lidar systems that can partially penetrate the canopy and
retrive the snow surface elevation.  Recent work has processed lidar point clouds
to resolve under canopy snowpack and validated the results against field
observations (Kostadinov et al., 2019; Safa et al., 2021).  Refinements to this
method offer promise for better resolving lidar returns from low canopy with the
snow surface (Piske et al., 2024) and creating datasets that can be used to train
models and improve other remote sensing snow products.



Numerical modeling of the snowpack allows simulating the complete state of the snowpack, including the SWE, at any spatiotemporal resolutions. Modern snowpack models of increasing complexity even represent the horizontal transport of the snow caused by wind and avalanches, and the interactions with forests (Mazzotti et al., 2020; Vionnet et al., 2021). However, numerical models often rely on adjustable parameters to represent different physical processes, whose transferability between different areas and model resolutions is usually complex, leading to uncertain simulations (Essery et al., 2013). In addition, these models rely on high resolution meteorological forcings, that are very challenging to generate and constrain, in part due to the lack of dense in situ observations. An alternative is to use meteorological downscaling techniques based on limited-area atmospheric models (Alonso-González et al., 2021; Sharma et al., 2023). However, the computational cost of regional atmospheric models increases significantly with finer resolution, with the current state of the art at the kilometer scale (Rasmussen et al., 2023). As such, dynamical downscaling is not yet a tractable option to couple with high and hyper resolution snowpack simulations. A partial, and very widespread, solution to this problem is to use simplified downscaling models that rely on different assumptions and/or empirical approximations to generate high resolution meteorological forcing fields. These may be predefined temperature and precipitation lapse-rates, or using empirical relations between the atmospheric variables and the underlying terrain (Fiddes & Gruber, 2014; Liston & Elder, 2006; Reynolds et al., 2023). Despite their simplicity, these more heuristic approaches may lead to a performance comparable with dynamically downscaled meteorological products (Alonso-González et al., 2023; Gutmann et al., 2012; Kruyt et al., 2022). Nonetheless, any (often considerable) remaining uncertainty in the forcing will, together with the uncertainty in the snow model structure and parameters, be propagated to the snowpack simulations, typically leading to simulations that differ significantly from reality (Krinner et al., 2018).

Data assimilation (DA) is the exercise of merging noisy observations with uncertain numerical models to exploit the strengths of both worlds (Evensen et al., 2022). Thanks to DA, It is possible to constrain model uncertainty using partial information from snowpack observations (Largeron et al., 2020). Although DA may not be as widespread in the snow sciences as in other disciplines, its use is becoming more common with a number of operational and experimental initiatives (e.g. Girotto et al., 2024; Mott et al., 2023). Using DA, it is possible to infer uncertain parameters to improve the simulations so as to better match the observations, providing an estimation of the model uncertainty. However, snow DA is still rarely used in forested areas due to the lack of reliable remote sensing observations of the snowpack under the canopy.



Canopy cover impedes the direct observation of the snowpack from space or
airborne sensors, which collaterally hampers the use of DA, and may even degrade
simulation outputs if implemented in its simplest form (Yatheendradas et al.,
2012). This is probably the reason that the majority of snow DA experiments have
been limited to arctic or alpine areas above the treeline, with only some
experiments approaching specifically the topic of snow DA in forested areas.
Smyth et al. (2022) tested the potential of a particle filter DA algorithm to improve
snowpack simulations generated by the Flexible Snow Model (FSM2) in the
presence of observations beneath the canopy. The results show that simulations
can be improved by assimilating data in snow models that consider canopy
interactions. However, the question of how to improve simulations of the
snowpack in case of a total occlusion of the snow view in certain regions of the
simulation domain (i.e. lack of local observations) remains unanswered. Pflug et
al., (2024) proposed a simplified three dimensional DA scheme to update the SWE
state variable at unobserved locations from remote observations in forest gaps
and tested their approach with a synthetic observing system simulation
experiment (OSSE). First, they used a one dimensional (purely temporal) Ensemble
Kalman Filter (EnKF) to update the cells where observations exist. In a second step
they updated the local unobserved pixels SWE using the ratio of the average
observations and average modeled SWE within a spatial window, generating a new
observation to be assimilated. Due to its simplicity, this heuristic procedure
succeeded in performing a promising synthetic assimilation experiment over a
very large area of North America at an affordable computational cost. Cho et al.
(2023) assimilated spatially coarse (5km2) airborne gamma ray based SWE
retrievals in forested environments, using a three-dimensional EnKF. These recent
works lay the foundations of snow data assimilation in forests, with great potential
to (i) improve snowpack simulations in forested watersheds, (ii) better understand
snow-forest processes, and (iii) identify shortcomings in snow-forest model
parameterizations. However, these previous works are based on necessarily
simplified approximations to limit the computational cost, synthetic experiments
or very coarse resolutions unable to capture the spatial variability present in
montane forests (Safa et al., 2021; Tennant et al., 2017). The emergence of new
technologies that allow the acquisition of snowpack observations at high and
hyper resolutions (Gascoin et al., 2024), make it necessary to adapt classical DA
techniques to maximize the value of the available information.
The interactions between the canopy and the snowpack behavior pose challenges
for inferring the snow mass beneath the canopy directly from nearby observed
locations in forest clearings, preventing simple interpolation techniques
(Dharmadasa et al., 2024) or DA techniques designed to update the model states





directly from the information obtained in nearby cells to work efficiently in this
context (Pflug et al., 2024). It is necessary to explore how the available information
can be transferred from the available observations in forest clearings to beneath
the canopy, where observations are typically either missing or highly uncertain. In
this work, we test a recently developed spatio-temporal snow DA methodology
(Alonso-González  et al., 2023), specifically designed to update distributed
snowpack simulations from spatially incomplete observations such as in a forest
environment where the information from remote sensors is mostly available in
forest clearings. We combine that information with a unique post-processed lidar
dataset that resolves the under-canopy snowpack explicitly (Kostadinov et al.,
2019; Piske et al., 2024) to validate the model.  The objective of this work is (i) to
explore the potential of lidar-derived real observations to update distributed
snowpack simulations at hyperresolution (10 m) scales in forest environments, and
(ii) to test different spatiotemporal DA configurations for estimating snow under
the canopy when only observations in forest gaps are available. Here we propose
different spatio-temporal DA configurations to propagate information under the
canopy where the observations are often not available.



## 2 Data and Methods

### 2.1 Observed snow depth maps, vegetation parameters and meteorological forcing

The experiments proposed in this work were developed in the Sagehen Creek forest (California, USA, Fig. 1). The observations consist of one airborne LiDAR derived snow depth map collected by the National Center for Airborne Laser Mapping on 21 March 2022 (Piske, 2022) and a snow off flight (Graup, 2021). From all the available areas, we have manually selected a domain of ~2x2 km that maximizes canopy heterogeneity and the observed snowpack data that are incomplete due to dense canopy cover, since pixels with potential vegetation-snowpack conflict were removed to increase the confidence in the snow depth data. The Sagehen Creek site was used to develop and test a new method of under canopy snow depth detection from airborne lidar (Kostadinov et al., 2019) that resolves a considerable amount of snow information beneath the canopy (Fig. 1). We use a slightly improved method to extract vegetation from the snow surface described in Piske (2022). Based on nearby SNOTEL at a similar elevation (SNOTEL Site: 539, Independence Camp, https://wcc.sc.egov.usda.gov/nwcc/site?sitenum=539, last accessed: 11-Nov-2024), the SWE was 43 cm on 21 March when lidar was collected compared to a maximum annual SWE of 48 cm on 9 March, 2022. The native spatial resolution of the lidar dataset was 1 m which was resampled to 10m for use in the DA analysis. The error variance of the observations was assumed to be $\sigma^2 = 0.01\,\text{m}^2$ at 10m resolution based on previous airborne LiDAR snow experiences that reported similar error metrics (Currier et al., 2019; Harpold et al., 2014; Mazzotti et al., 2019; Painter et al., 2016). Future initiatives may benefit from more sophisticated error models. In addition to the snow depth observations, different vegetation parameters were computed from the three-dimensional lidar data, including vegetation height, the Vegetation area index and the in forest sky view factor based on methods described in Broxton et al., (2015) and Broxton et al. (2021). This dataset was segmented into grid cells in forest clearings (to be assimilated) and canopy-covered cells (to be used as independent validation) based on this vegetation information. The meteorological forcing was generated using MicroMet (Liston & Elder, 2006) forced by the ERA5 atmospheric reanalysis (Hersbach et al., 2020). The meteorological fields were downscaled to the same geometry of the observations using a LiDAR based digital elevation model (Sourp et al., 2024). The precipitation partitioning was estimated using the psychrometric parameterization scheme proposed by Harder & Pomeroy (2013).



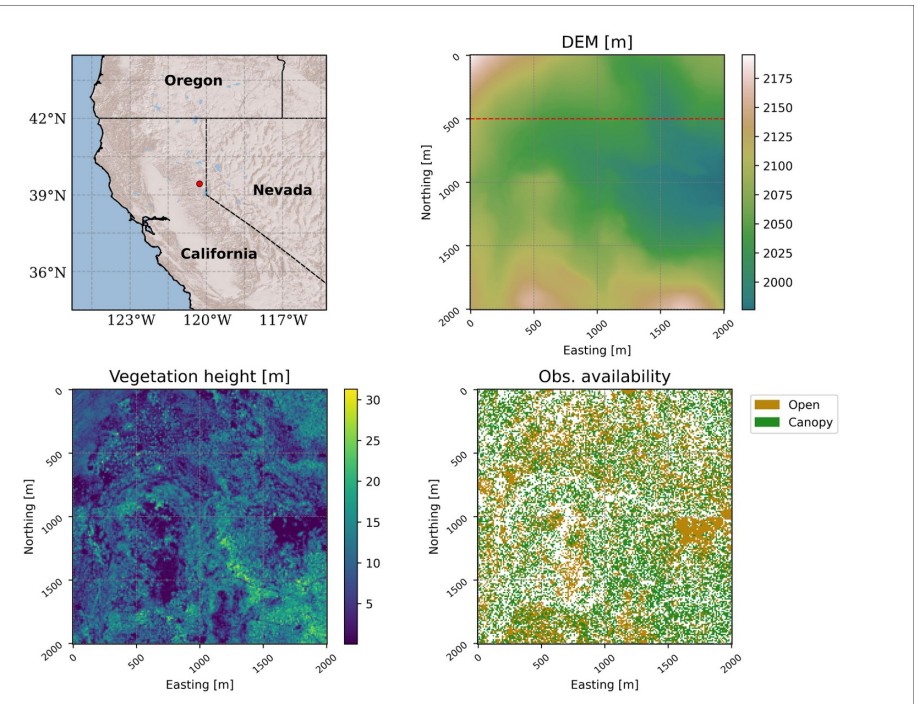

Figure 1. Localization map, Digital elevation model, vegetation height and available observations
(with its segmentation between canopy covered data used for validation or forest gaps to be
assimilated). The red transect in the digital elevation map indicates the location of the profile used
later for validation.
## 2.2 Data assimilation and computational setup



All DA experiments presented in this work were developed using the Multiple
Snow data Assimilation (MuSA) system (Alonso-González et al., 2022). MuSA is an
open-source DA toolbox designed primarily as a python wrapper around the
Flexible Snow Model (FSM2, Essery et al. (2024), but now providing support for
other numerical models as well while not necessarily being limited to snowpack
models. MuSA provides support to different DA algorithms, and simplifies the
implementation of new ones thanks to its modular design. In this work, the FSM2
model was chosen due to its already coupled canopy module that required only
minimal modifications of the original MuSA code to be activated. MuSA, and
therefore FSM2, was forced by the Micromet outputs, and provided with the
aforementioned lidar-derived vegetation parameter maps. It should be noted that
although in this work we focus on the MuSA snow depth outputs (as this is what
we can validate) posterior simulations include the full state vector of FSM2,
including SWE.
The spatio-temporal DA scheme is described in Alonso-González et al. (2023), and
therefore we only briefly introduce some key points here, its configuration, and
the new modifications implemented to improve its performance for the new
problem at hand. We only infer meteorological correction parameters and not
model states, leading to physically consistent (in terms of FSM2) simulations of the
modeled snow state across the snow season. As mentioned above, snowpack in
the forest gaps shows a different behavior than beneath the trees (Varhola et al.,
2010), so trying to infer canopy-occluded states directly from the information we
can obtain in the gaps could also introduce artifacts in the simulations. Crucially
the forcing perturbations will also be modified by the canopy scheme in FSM2, so
even if the above canopy forcing is constrained to be similar for neighboring cells
the forcing that the below canopy snowpack experiences will be different thanks to
the model physics.





As the first step in our workflow, we generated an ensemble of 100 FSM2
simulations by randomly drawing stationary (i.e. constant across the water year)
spatially correlated prior parameters to perturb the meteorological forcing,
particularly the precipitation and 2m air fields. The choice of perturbing only
precipitation and temperature was motivated by previous successful experiments
with a similar setup, albeit in non-forested environments (Alonso-González et al.,
2023; Alonso-González et al., 2022). Herein, the prior probability distributions that
we sampled using a random number generator were: a normal (additive
parameter) for the temperature bias and lognormal (multiplicative parameter) for
the precipitation scaling. These prior distributions were defined by: its mean ($\mu = 0$)
and standard deviation ($\sigma = 1$) in the case of the temperature and by the mean and
standard deviation ($\mu = 0$, $\sigma = 0.4$) of the *underlying* normal distribution in the case
of the precipitation. The latter results in log-normally distributed prior
multiplicative precipitation scaling parameters in the physical space whose median
is ~1. The objective of the algorithm is to update these parameters by assimilating
observations to directly correct the temperature and precipitation fields and
indirectly update the corresponding snowpack states
For this purpose we have used a deterministic ensemble Kalman filter (DEnKF)-
based algorithm in iterative smoother mode, namely the Deterministic Ensemble
Smoother (DES, Sakov & Oke, 2008) with multiple data assimilation (DES-MDA,
Emerick, 2018). In this DES-MDA scheme, the update proceeds in two steps for
each grid cell $i = 1, \ldots, N_g$ and $\ell = 0, \ldots, (N_a - 1)$ MDA iteration. Firstly, the $N_p \times 1$
updated ensemble mean parameter column vector $\overline{\boldsymbol{\theta}}_{\ell+1}^{(i)}$ is obtained using a Kalman
analysis equation of the form

$$\overline{\boldsymbol{\theta}}_{\ell+1}^{(i)} = \overline{\boldsymbol{\theta}}_{\ell}^{(i)} + \mathbf{K}_{\ell}^{(i)} \left[ \mathbf{y}^{(i)} - \overline{\mathbf{y}}_{\ell}^{(i)} \right]$$





where $\overline{\theta}_\ell^{(i)}$ is the $N_p \times 1$ ensemble mean parameter column vector from the current
(prior for $\ell = 0$) iteration, the $N_p \times N_o^{(i)}$ matrix $\mathbf{K}_\ell^{(i)}$ is a localized and inflated
ensemble Kalman gain computed using ensemble covariances and the observation
error covariance, $\mathbf{y}^{(i)}$ is the $N_o^{(i)} \times 1$ *local* observation vector containing available
local observations that are within a yet to be defined distance-based neighborhood
$d < 2c$ (see the GC function below) of grid cell $i$, and the $N_o^{(i)} \times 1$ vector $\overline{\mathbf{y}}_\ell^{(i)}$ contains
the corresponding local ensemble mean predicted (i.e. modeled) observations
from the last iteration obtained at neighboring grid cells. We refer to Alonso-
Gonzalez et al. (2022) for the full form of the ensemble Kalman gain matrix in
particular and a more detailed overview of the implementation of spatio-temporal
DA using the DES-MDA in MuSA in general. Secondly, the $N_p \times N_e$ matrix $\mathbf{\Theta}_{\ell+1}^{(i)\prime}$
containing the updated ensemble of parameter vector anomalies (from the mean)
is obtained a modified Kalman analysis equation of the form

$$\mathbf{\Theta}_{\ell+1}^{(i)\prime} = \mathbf{\Theta}_\ell^{(i)\prime} - 0.5\mathbf{K}_\ell^{(i)} \left[ \mathbf{y}^{(i)} \mathbf{1}_{N_o^{(i)}}^{\mathrm{T}} - \widehat{\mathbf{Y}}_\ell^{(i)} \right]$$

where $\mathbf{\Theta}_\ell^{(i)\prime}$ is the $N_p \times N_e$ matrix containing the ensemble of parameter vector
anomalies from the current iteration, $\mathbf{1}_{N_o^{(i)}}^{\mathrm{T}}$ is a $1 \times N_o^{(i)}$ row vector of ones, and $\widehat{\mathbf{Y}}_\ell^{(i)}$
is the $N_o^{(i)} \times N_e$ matrix of predicted observations from the current iteration. Once
the mean and anomaly update steps have been carried out, the $N_p \times N_e$ matrix
$\mathbf{\Theta}_{\ell+1}^{(i)}$ (*without* the prime) containing the updated ensemble of parameter vectors is
obtained through the matrix sum

$$\mathbf{\Theta}_{\ell+1}^{(i)} = \overline{\theta}_{\ell+1}^{(i)} \mathbf{1}_{N_p}^{\mathrm{T}} + \mathbf{\Theta}_{\ell+1}^{(i)\prime}$$

where $\mathbf{1}_{N_p}^{\mathrm{T}}$ is a $1 \times N_p$ row vector of ones. Unlike the classic stochastic (perturbed
observation) ensemble Kalman scheme, this deterministic ensemble Kalman
scheme is less overconfident thanks to built-in model covariance inflation and also
avoids the need to factorize the observation error covariance that can be costly in
spatio-temporal problems (Emerick, 2018). In the loop over iterations above we
implicitly rerun the forward model, FSM2 in this case, with the updated parameter
values to generate an updated ensemble of hidden snowpack states including the
predicted snow depth observations to be assimilated.



The Gaussian assumptions inherent in this ensemble Kalman method make it
more robust against ensemble collapse (where a single member carries all the
posterior probability) than particle methods which are more widely used for snow
DA (Alonso-Gonzalez et al., 2022). In particular, we have used an iterative version
of DES, that performs the update of the parameters in multiple data assimilation
(MDA) steps, creating the DES-MDA used here (Emerick, 2018). The MDA is a form
of likelihood tempering (Murphy, 2023)that helps relax the undesirable effects of
the linear assumption inherent to EnKF based algorithms. In nonlinear DA
problems such as the one tackled here, previous work has shown that these MDA
iterations lead to significant improvement of the results compared with non-
iterative versions of the algorithm(Aalstad et al., 2018; Alonso-González et al.,
2022). In this work, based on previous studies (Alonso-González et al., 2022 and
references therein), the number of iterations was fixed to 4. To accommodate the
Gaussian assumption, we employed analytical Gaussian anamorphosis (Bertino et
al., 2003) to log transform the precipitation parameter distribution to a normal
distribution and perform the update in Gaussian space. After the update, the
parameters are mapped back to the model space using the exponential function
before generating the new ensembles.



The spatial propagation of information may happen through two main
mechanisms in the DES-MDA: observation error correlations or prior correlations
(van Leeuwen 2019). Since observation error correlations are more challenging to
specify and arguably less general than prior correlations, we will focus only on the
latter. A key component of the scheme is to draw random prior parameters for
each cell that are correlated with other cells in the domain, reflecting similarities
among different regions of the simulation domain. In the general DA literature,
this is typically done by computing the pairwise geographic (Euclidean) distance to
map the proximity of the cells. The pairwise distance matrix is then used to
generate a covariance matrix. In this work we have used the 5th-order piecewise
rational function proposed by Gaspari and Cohn (GC) (Gaspari & Cohn, 1999), as is
often done in DA to generate and localize the covariance matrix. The GC
localization function depends on an important hyperparameter, the correlation
length scale, that in practice controls how far information can be transfered
spatially. Crucially, this length scale willaffect both the posterior results and the
computational cost since a larger length scale results in a greater number of
neighbors with non-zero correlation. The GC function defines a distance-based
correlation                             as                      follows                        :

$$
\rho\left(d,c\right) = \begin{cases} -\frac{1}{4}\left(\frac{d}{c}\right)^5 + \frac{1}{2}\left(\frac{d}{c}\right)^4 + \frac{5}{8}\left(\frac{d}{c}\right)^3 - \frac{5}{3}\left(\frac{d}{c}\right)^2 + 1\,, & \text{for } 0 \leq \left(\frac{d}{c}\right) \leq 1\,, \\ \frac{1}{12}\left(\frac{d}{c}\right)^5 - \frac{1}{2}\left(\frac{d}{c}\right)^4 + \frac{5}{8}\left(\frac{d}{c}\right)^3 + \frac{5}{3}\left(\frac{d}{c}\right)^2 - 5\left(\frac{d}{c}\right) + 4 - \frac{2}{3}\left(\frac{d}{c}\right)^{-1}\,, & \text{for } 1 \leq \frac{d}{c} \leq 2\,, \\ 0\,, & \text{for } \frac{d}{c} > 2\,. \end{cases}
$$



where, $d$ is the pairwise distance between cells and $c$ is the correlation length
scale. This function is used for localization, with two important roles: first, it
reduces spurious long range correlations that arise due to the limited size of the
ensemble (Morzfeld & Hodyss, 2023), and second, to save considerable
computational costs since relatively distant locations can be ignored when
updating a particular cell. . Note that without localization, the spatio-temporal DA
problem would essentially be intractable, especially in this context with a relatively
large domain and a high spatial density of observations. In addition to ensemble
collapse, this is another motivation for using the ensemble Kalman method over
particle techniques here, since more developed localization methods exist for the
former (Evensen 2022). Despite being the typical spatial snow DA configuration
(e.g. De Lannoy et al., 2012; Magnusson et al., 2014) and references herein), there
is no reason to restrict the distance mapping to the geographic (northing and
easting dimensions) space, since an arbitrary number of dimensions can be used
to define a feature space and generate the distance matrix. It is widely
acknowledged that snowpack redistribution is strongly dominated by the
topographic characteristics of the terrain, such as concavity, slope, and elevation
as well as vegetation parameters (e.g. Dharmadasa et al., 2023; Essery & Pomeroy,
2004; Revuelto et al., 2014; Zheng et al., 2019). In the context of snow DA, it is
possible to map the similarities between cells using a multidimensional feature
space of topographical (or any other) dimensions. The only two considerations to
be taken into account are that these feature dimensions may have different units, and
that they can be potentially correlated. This may generate a space of non-
orthogonal dimensions where using the Euclidean distance directly may lead to a
spurious similarity mapping (Curriero, 2006). It is possible to overcome these
issues by using the Mahalanobis distance, a generalization of the Euclidean
distance that includes a covariance-based normalization attempting to address
these two problems in a single step. Alternatively, it may be possible to generate
other spaces using synthetic transformed orthogonal dimensions in a potentially
lower dimensional space from the previously scaled topographical dimensions
using a principal components analysis or multidimensional scaling approaches
(e.g. Aversano et al., 2019; Murphy et al., 2015), and compute the pairwise
Euclidean distance matrix in the new synthetic space.



Whichever approach is used to define the space that enables information to be
spread, it is necessary to generate a pairwise distance matrix to compute a prior
covariance matrix. The previous version of MuSA generated the complete distance
matrix, which is highly memory and time intensive with poor scalability. The
reason for this is that the computational cost and the size of the matrix scales
quadratically with the number of cells, further complicating subsequent linear
algebra operations. However, it is not necessary to compute the full distance
matrix since localization ensures that long distances will be ignored in the analysis
as the corresponding elements in the covariance matrix will be 0 beyond a certain
distance that is controlled by the GC length scale hyperparameter. This makes the
distance and the subsequent covariance matrix very sparse, opening new
possibilities to make the otherwise expensive prior sampling more tractable. As
such, in MuSA we have now implemented the capability of mapping the distances
using a k-dimensional tree (k-d tree) space-partitioning data structure, as
implemented in the SciPy python module (Virtanen et al., 2020). This allows MuSA
to ignore all distances beyond the GC hyperparameter value, generating a sparse
distance matrix. Unfortunately only Minkowski metrics (which includes the
Euclidean distance) are available so far with the k-d tree implementation. As such,
this method is not compatible with Mahalanobis spaces in the current MuSA
version, and therefore it was not used for all the experiments proposed here. In
addition, we have implemented the capability of computing the distance matrix
cell by cell, which has proven to be very memory efficient with a very manageable
loss of efficiency that is compatible with Mahalanobis, or any other, distance
metric. Since the distance matrix, and the generated prior covariance matrix, are
very sparse, we have now migrated most MuSA linear algebra routines to the
SciPy.sparse module. This allows for the use of sparse linear algebra, enabling us
to sample even in very large domains while, depending on the GC
hyperparameter, maintaining an affordable computational cost. All these
modifications are included in a new MuSA version (v2.2), compatible with the use
of arbitrary masks, even non-contiguous ones within the same simulation domain,
indicating over which cells to perform the analysis. This allows simulations to be
performed only in the areas of interest such as. above a certain elevation or within
a certain complex basin geometry), while still performing spatio-temporal
assimilation by propagating the information between the selected cells at a
considerably reduced computational cost.





The last step of the prior sampling requires approximating the square root of the
covariance matrix via Cholesky factorization. As noticed by previous research
(Alonso-González et al., 2023; Curriero, 2006), the use of non-Euclidean distances
(e.g. using the Mahalanobis distance) leads easily to non-positive definite
covariance matrices, making it impossible to compute the Cholesky factor. We
have increased the numerical stability of the prior sampling in MuSA by
regularizing the prior covariance matrix, adding small values to the elements of its
diagonal. These diagonal elements are increased iteratively up to a limit defined by
the user (from 1e-6 to a maximum of 0.1 in this study), following a technique
known as jitter as is commonly done in the Gaussian Process machine learning
community (Neal, 1999; Rasmussen & Williams, 2005). The remaining steps,
including the DES-MDA update itself, remain the same as in the previous version of
MuSA, despite a few minor updates with the intention of improving the I/O
performance by optimizing the compression routines. All these modifications are
packed as a new version, whose code has been released together with this work
(Alonso-González et al., 2024).
## 2.3 Experimental design
We propose different experiments to evaluate the potential of ensemble-based
data assimilation techniques to update hyperresolution simulations in forest
environments. First, as a reference, we generated a deterministic reference run
without any DA for comparison with the updated simulations. Then, different
experiments were developed in an effort to find a MuSA configuration that is able
to exploit dispersed hyperresolution information in forested terrain. Here we are
not aiming to find a generalistic optimal configuration, since each specific case will
require a different configuration, depending on the resolution of the simulations,
the spatial density of the observations, the domain, and the availability of
computational resources. We propose 3 different information propagation
schemes, and two different GC hyperparameters for each, leading to 6 different
simulations:
• Using Euclidean distances in the geographical space. We developed two
different simulations where the Euclidean distance over the northing and
easting dimensions is used to map the similarities among cells, using the
values of 50 (Eu50) and 100 (Eu100) m for the GC hyperparameter.





- Using the Mahalanobis distance in a topographical space. Here, we propose two experiments where in addition to northing and easting, we included elevation, the Topographic Position index, the Diurnal Anisotropic Heat index and the slope to define a topographical space. Since we have separated the data beneath the canopy and in the forest gaps, using them for assimilation and validation data, it is not instructive to include dimensions based on vegetation parameters. In fact, due to the GC function, it might even prevent the information transfer towards the canopy covered cells. The open cells that are geographically (or topographically) distant, and nearby geographically (or topographically) cells under the canopy, would be equally far away in Mahalanobis distance from a given open observed cell in that hypothetical space including vegetation parameters. The distances were computed using the Mahalanobis distance (Ma), and the GC hyperparameters tested were 0.5 (Ma0.5) and 1 (Ma1).
- Using Euclidean distances in a synthetic topographical space. Here we included a PCA (after z-score standardization) analysis over the topographical space to generate an orthogonal space that ensures a positive definite covariance matrix by sorting the cells prior to computing the covariance matrix. This saves significant computational cost since it allows for distance mapping using the new k-d tree implementation. The number of principal components was selected automatically using the algorithm proposed by Minka (2000), which in practice resulted in 5 components. The GC hyperparameters tested were 0.5 (PCA0.5) and 1 (PCA1).

For each of the experiments, we have computed the cell wise Continuous Ranked Probability Score (CRPS, Hersbach, 2000), a generalization of the mean absolute error for probabilistic simulations:

$$\mathrm{CRPS}(F, x^\star) = \int_{\mathbb{R}} \left[ F(x) - H(x - x^\star) \right]^2 \, \mathrm{d}x$$

Where $F(x)$ is the predicted cumulative distribution function of the snowpack state variable $x$ to be evaluated, $x^\star$ is the reference (ground truth) value for the state obtained from observations, and $H(x - x^\star)$ is the Heaviside function resulting in $1$ if $x \geq x^\star$ and $0$ otherwise. We have used a normal approximation of the posterior snow depth distribution defined from the posterior mean and standard deviation derived from the ensemble together with the observations to compute the cell by cell mean CRPS and standard deviation (SD). We have also computed the spatial




bias, which is the mean error of all cells used for validation, where error is the difference between the posterior mean and the observations. In addition, we computed the correlation (R) and root mean square error (RMSE) between the posterior mean and observations across the domain. To evaluate the spatial patterns of each of the experiments, we calculated the variograms of each simulation. To quantify how far the variogram curves are from the one obtained from the observations under the forest canopy, we used the discrete Frechet distance (FrDist) as an indicator of similarity between the variogram curves.



## 3 Results

### 3.1 Validation metrics of the reference run and DA experiments

Compared with the deterministic reference simulation, both the Euclidean (Eu) and Mahalanobis (Ma) experiments improved the quantitative error metrics considerably (Table 1). The marked improvement in R (from R = 0.1 to R ~0.8 on average for all the Eu and Ma experiments) is especially notable, and, combined with the lower Frechet distance values (FrDist = 0.29 for the reference, while FrDist = 0.005 on average for the Eu and Ma experiments), indicates a significant improvement of the spatial patterns of the simulation. RMSE values also improved significantly (RMSE improvement ~30%). The bias remained lower and close to zero (bias mean = -0.07 m) for the reference simulation compared with the Eu and Ma experiments (bias mean ~ 0.13 m), suggesting a slight overestimation of the snow mass in the updated simulations. However, the RMSE in the reference run (RMSE = 0.32) compared with the Ma and Eu experiments (RMSE = 0.2) suggest many cells in the reference run exhibit higher errors than the ones of the Eu and Ma experiments. The CRPS, which is the only uncertainty-aware metric considered that accounts for both the precision and accuracy of the ensemble, showed lower values for the Eu50 (CRPS = 0.12), but followed closely by the other experiments, except the PCA0.5 and PCA1.

Unfortunately, despite the convenience of using a PCA preprocessing step, the experiments using PCA exhibited only a slight improvement in some metrics while degrading other indicators. In particular, they exhibited a slight improvement in the correlation values (R=0.20 and 0.46 for PCA0.5 and PCA1 respectively), while all other metrics were similar to the reference (e.g. bias), with a FrDist being equivalent or significantly degraded relative to the reference for the PCA0.5 (FrDist = 0.021) and PCA1 (FrDist = 0.046), respectively. This suggests not only that absolute error metrics were not improved, but even that spatial patterns were not adequately simulated with the PCA approach.

Table 1: Validation metrics of the experiments

| Exp. | RMSE | R | Bias | CRPS [mean(+/- SD)] | FrDist |
|---|---|---|---|---|---|
| Ref. | 0.32 | 0.10 | -0.07 | - | 0.029 |
| Eu50 | **0.20** | 0.84 | 0.12 | **0.12**      (+/- | 0.006 |



| | | | | **0.09)** | |
|---|---|---|---|---|---|
| Eu100 | 0.22 | **0.85** | 0.15 | 0.13 (+/- 0.12) | 0.009 |
| Ma0.5 | 0.22 | 0.76 | 0.10 | 0.14 (+/- 0.10) | **0.003** |
| Ma1 | 0.24 | 0.81 | 0.16 | 0.15 (+/- 0.12) | **0.003** |
| PCA0.5 | 0.33 | 0.20 | **-0.03** | 0.19 (+/- 0.13) | 0.021 |
| PCA1 | 0.33 | 0.46 | 0.08 | 0.21 (+/- 0.18) | 0.046 |



Among the Eu experiments, Eu50 exhibited slightly better or similar error metrics
than the Eu100. However, the differences were minimal, suggesting there is
flexibility in choosing the GC hyperparameters, in this case at least, in terms of
validation metrics. A similar conclusion can be drawn from the validation metrics
of the Ma experiments, where there was not a clearly superior simulation.
Similarly, Eu and Ma yielded comparable performance according to these error
metrics. However, the FrDist metric was consistently lower in the Ma experiments
compared with the Eu experiments, suggesting a better representation of the
spatial patterns, while the remaining error metrics were slightly better or similar
for the Eu experiments. This superior performance in representing the spatial
patterns was evident in the snow depth semivariograms of the experiments (Fig.2),
where Ma experiments exhibited a semivariance much closer to the observations,
even reproducing accurately the nugget effect exhibited by the observations,
suggesting a better representation of the small scale patterns. In any case, the
variograms of the Eu and Ma experiments exhibit a closer shape to the one
obtained from the observations, compared with the one  obtained in the reference
run, which is nearly flat.

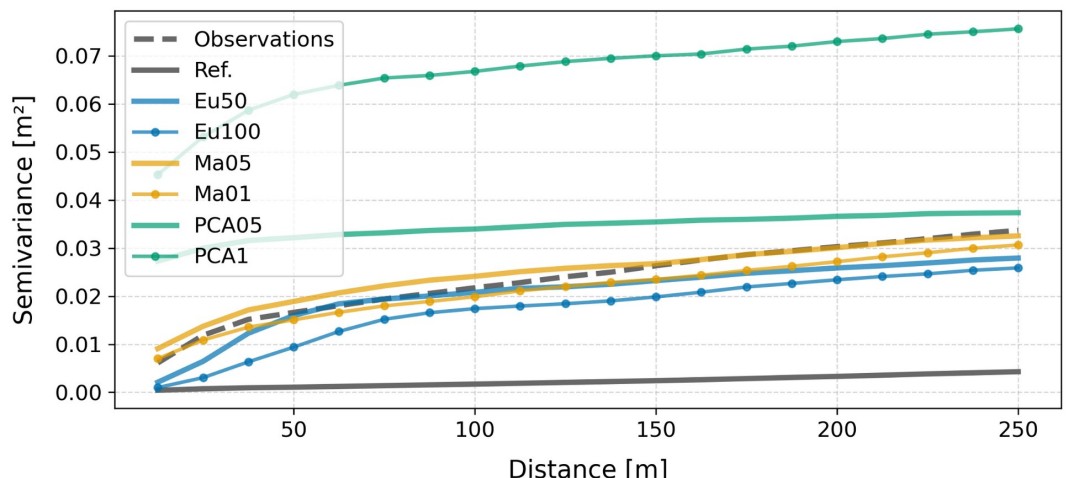

Figure 2: Snow depth spatial semivariance derived from the lidar-derived observations, the
reference run and different experiments.



When examining the distributed posterior mean simulations, these considerations
about the spatial patterns become evident (Fig. 3). First, there was a very limited
spatial variability in the deterministic reference run, as reflected quantitatively by
the Frechet distance and qualitatively by the variograms. Among the Eu50 and
Ma0.5 posterior maps, there is a clear difference in its snow depth spatial patterns.
While the large scale patterns were similar in both simulations, and close to the
observations, the small scale patterns were different. In Eu50 small scale patterns
of the posterior mean were clearly affected by the shape of the GC function, since
the blurrier horizontal patterns are reminiscent of the Gaussian-like shape of this
function. On the other hand, Ma0.5 small scale patterns, which do not depend
solely on geographic distance, are considerably more intricate, which also explains
the lower FrDist error metric.

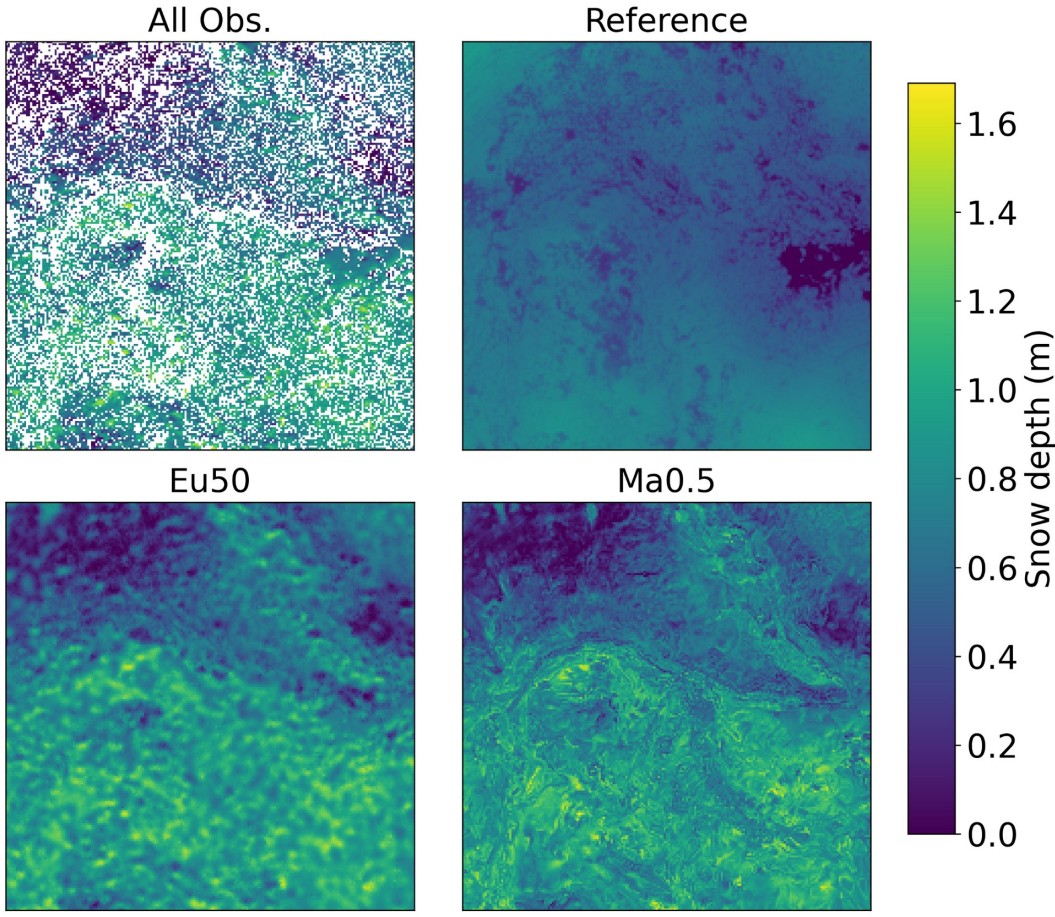





Figure 3: Distributed snow depth observations, reference simulation and posterior mean
simulations of the Eu50 and Ma0.5 experiments
While both in Ma0.5 and Eu50 point scale comparison with observations show a
similar overall R metric and distribution, it is worth noting the differences shown in
Fig.4. In Ma0.5, the cells with local observations (i.e. the cells in the forest gaps,
which include assimilated information) exhibit slightly larger residuals (R = 0.99
and R=0.97 for Eu50 and Ma0.5 respectively). These differences suggest that the
influence of the GC hyperparameter makes both schemes not fully comparable.
This is a consequence of the varying number of observations used to update the
parameters of each cell that differ for each experiment, depending on how much
space falls within the correlation length scale of the GC function in each case.



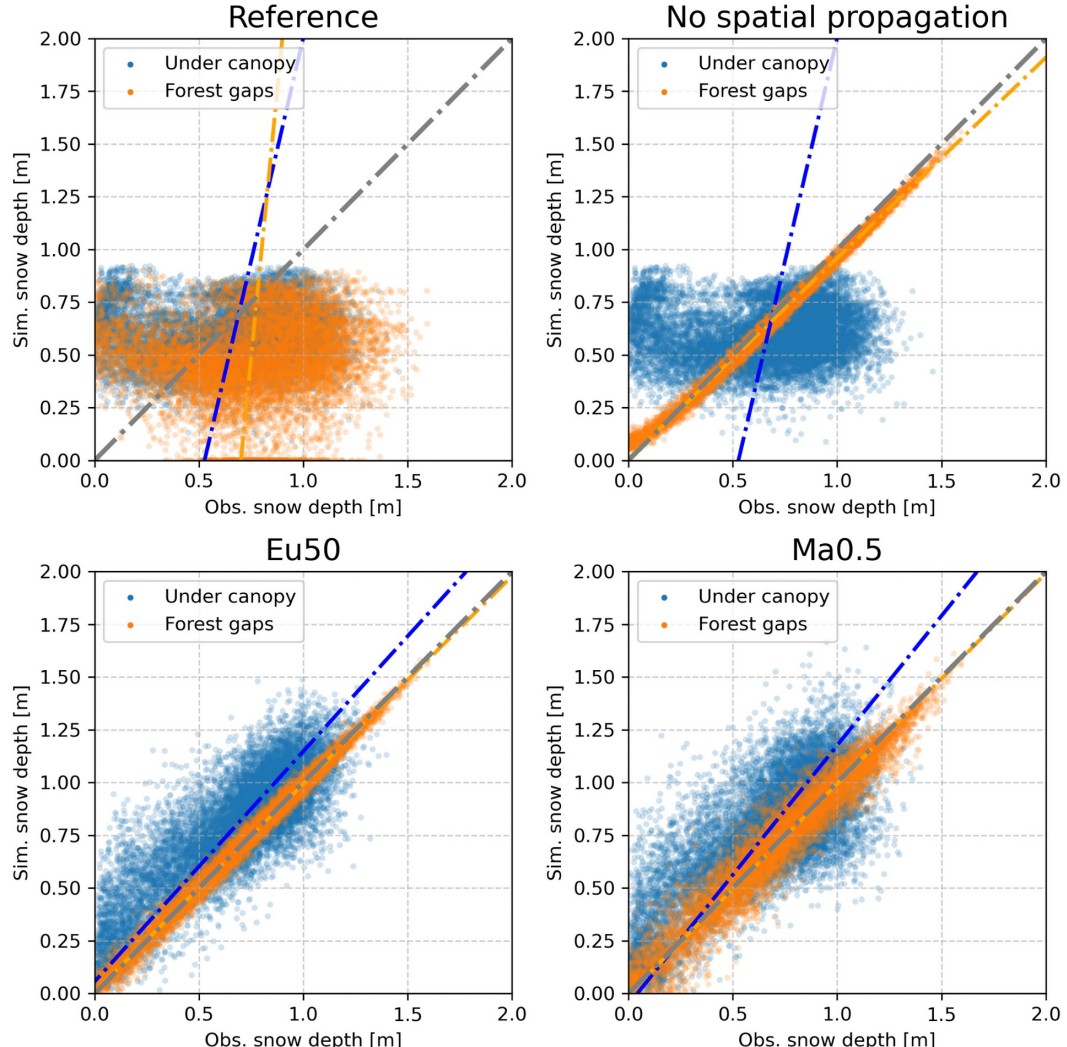

Figure 4: Scatterplot based comparison of the under canopy (withheld) and forest gaps
(assimilated) observations, with the reference simulation, and the posterior mean of a data
assimilation experiment without and with the spatial propagation of the information enabled
(experiments Eu50 and Ma0.5)




However, these error metrics should be taken with care. Most of them (except
CRPS) used the posterior mean as an optimal point estimate  of the updated
simulation. This assumption was adopted for simplicity but may compromise the
interpretation of the results. Posterior simulations are not deterministic
simulations and come with an  uncertainty estimate inherent in the posterior
ensemble. To investigate this issue, we extracted a longitudinal profile along the
easting dimension, including both the deterministic reference simulation and the
posterior mean, but for the latter we now included the associated uncertainty
represented by +/-1 posterior standard deviation (which accounts for
approximately 68% of the posterior probability, Fig. 5). In addition, we included a
representation of the observations obtained both beneath the canopy and in
forest gaps. The profile highlights the differences of using the GC function in the
Euclidean or topographic space, with Eu exhibiting a much smoother surface
compared with the sharper Ma profile. Both profiles exhibited a similar
performance if we account for the uncertainty. In terms of the posterior mean,
Ma0.5 was able to accurately capture snow depth in large areas beneath the
canopy (e.g. Fig.5 from 1000 to 1250), while maintaining most of the observations
in at least the range of its standard deviation. Both Eu50 and Ma0.5 improved the
reference run, which exhibited an evident underestimation and lack of
heterogeneity along this profile, with only a few observations approaching the
simulated reference values.

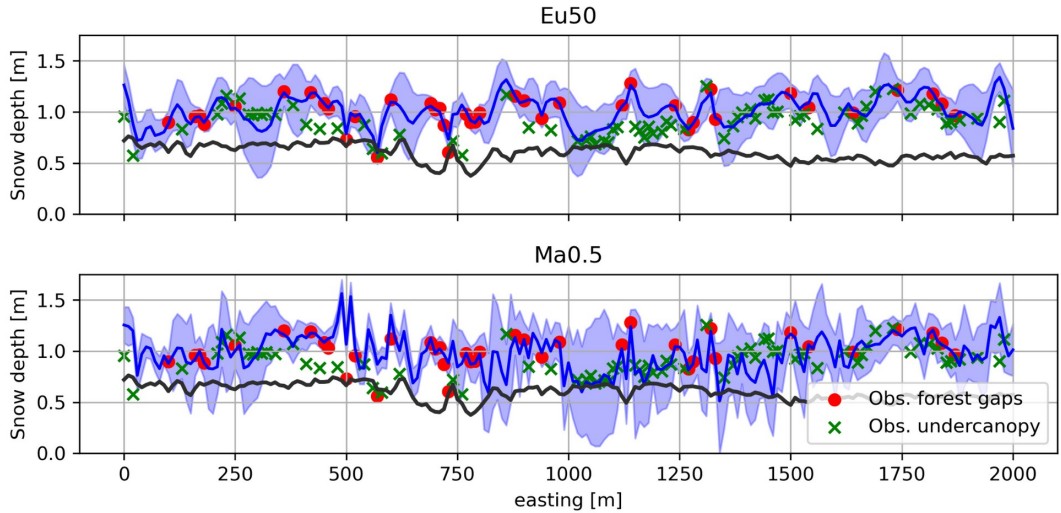




Figure 5: Snow depth profile showing the match between the reference run (black line), the Eu50
and Ma0.5 experiments and the observations for the horizontal profile delineated by the red line
shown in Figure 1. The dark blue line is the posterior mean and the shaded area the posterior
standard deviation.
Although the aim of the present work is to explore how to propagate the
information spatially, it is tempting to analyze the posterior distribution of the
parameters (Fig. 6). On average for all cells, using the experiment Ma0.5 as a
reference, the multiplicative precipitation parameter was 1.06 (+/- 0.30) and the
additive temperature parameters was -0.04 (+/-0.73).

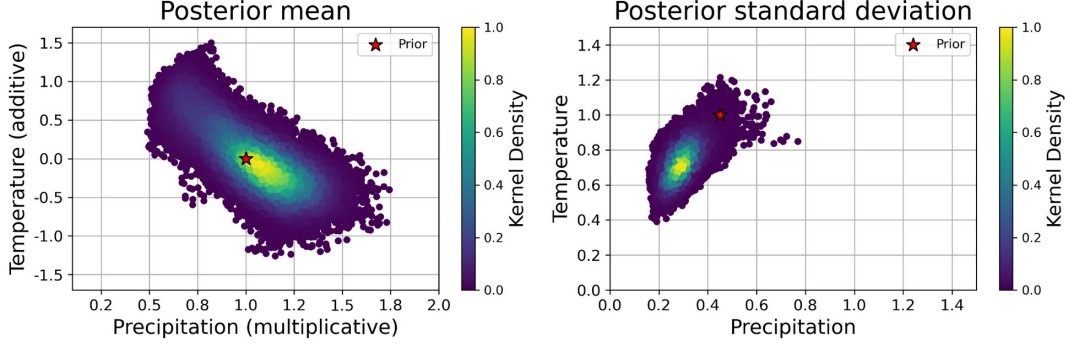

Figure 6: Posterior distributions of perturbation parameters in the model space for the Ma0.5
experiment, each point represents a grid cell.
**4 Discussion**





The results shown here demonstrate the potential of ensemble-based DA
experiments to improve hyper resolution snowpack simulations in forested terrain,
by updating the canopy covered cells from information retrieved in clearings.
Recall that the DA schemes proposed herein are theoretically independent of the
underlying numerical model,meteorological forcing or site. As such, in practice any
other snow or land surface model forced by meteorological data generated by any
downscaling tool at any geographical location may benefit from the proposed
techniques. The aim of this work is not to perform the best possible simulation,
but to explore whether it is possible to improve snowpack simulations in forested
areas by means of DA. Future initiatives may choose to explore the added value of
including additional forcing corrections or internal model parameters in the
parameter vector since there is, in theory, not any particular limitation on this
provided that a large enough ensemble is computationally feasible.
All experiments were performed using the Centre National D'Etudes Spatiales
(CNES) supercomputing infrastructure. For reference, the Ma0.5 experiment took
one day and eight hours to complete, using 6 nodes with 10 CPUs each to solve the
40401 cells (201 cells in each geographical direction) that compose the domain
using the aforementioned DA scheme. This estimate of computational cost, which
could be considered very affordable, especially given the iterative nature of the
assimilation algorithm and the relatively low number of processors involved,
should be treated with some caution. The computational time varied significantly
between experiments, as in practice the I/O increases with the GC
hyperparameter, which effectively defines a search radius. In addition, MuSA
benefits from distributed systems that share I/O bottlenecks among their nodes,
so the computational scheme can also be very relevant. On the other hand, other
DA experiments with a lower density of observations will see their computational
cost dramatically reduced, independent of the GC hyperparameter.



Most of the DA configurations managed to improve the posterior simulations
compared with the deterministic reference simulation, with different
configurations showing similar error metrics. However, the PCA based
experiments, despite their desirability given the orthogonal properties of the
synthetic coordinate system, did not perform as expected. We hypothesize that the
limitations found may come from the fact that the new set of coordinates do not
explicitly preserve the Cartesian northing and easting information by mixing them
with other dimensions, relaxing the relations between nearby cells in the Euclidean
space (Davis & Curriero, 2019). However, the same could be said when using the
Mahalanobis distance, but the performance of the Ma experiments was clearly
superior compared to the PCA ones. A potential reason may be the fact that, to
ease the positive-definiteness of the PCA-based covariance matrix by sorting the
cells in a lower dimensional space, we used the Minka algorithm to reduce the
dimensionality of the synthetic coordinate system. This dimension reduction
comes in practice with a loss of information. However, this is very unlikely, since in
practice it resulted in only one dimension being removed, which represented a
very low percentage of the total variance of the system. This requires further
research to fully understand how the information can be effectively propagated in
different spaces. A potential future approach may be the use of multidimensional
scaling techniques, instead of PCA, that have shown previous success in
geostatistics (R. R. Murphy et al., 2015). The challenges previously encountered in
generating non-positive definite covariance matrices have been substantially
mitigated. Previous research has proposed to enforce positive definiteness in
covariance matrices by using (potentially iterative) methods based on
eigendecomposition, to make any negative eigenvalues of the covariance matrix
become nonnegative (e.g. Davis & Curriero, 2019 and references herein), which
imposed a considerable computational burden, particularly for large matrices.
However, regularizing the covariance matrix with the introduction of the jitter
technique (where small values are iteratively added to the diagonal) has proven to
be both highly effective and computationally efficient. Whether the results of prior
sampling differ significantly between these two approaches to regularize the
covariance matrix remains an open question for future investigation.



The fact that in these experiments we update meteorological correction
parameters only, and not snowpack states, allows the numerical model to resolve
the snow-canopy interactions. This prevents the posterior simulations to be
degraded by the fact that in reality the snowpack beneath the canopy behaves
differently than in open terrain (Pflug et al., 2024; Varhola et al., 2010), by updating
only parts of the simulation that we assume to be similar independently of the
canopy cover (such as the precipitation or temperature), and letting the model to
resolve the parts that can't be constrained (such as snow states), due to the lack of
information. Since the main objective of this experiment was to explore how the
information can be propagated effectively from clearings towards the canopy
covered cells, we split the observation dataset in two, keeping the cells beneath
the canopy for validation. This has not allowed us to include vegetation
parameters in the distance mapping of the Ma experiments, as the cells inside and
outside the forest would have been too far away in Mahalanobis space, and
therefore due to the localization, the information would not have been transmitted
from the clearings towards the sub-canopy. Some vegetation model parameters
could have been included in the inference, but because the information is located
in the forest gaps, they could not have been constrained. However, given the
success of the experiments, future research would benefit from assimilating data
also in canopy-covered cells, if a proper error model is developed. State of the art
remote sensing techniques are able to retrieve at least a partial information of the
snowpack in forested terrain (Mazzotti et al., 2019), or even snow cover
information from satellites (Xiao et al., 2022). This may be used not just to further
improve the posterior simulations but as a tool to infer internal model parameters
spotting weakness in canopy/snow models or their parameters. It should be noted
that these spatio-temporal techniques are compatible with joint DA initiatives,
where more than one type of observation is assimilated into the same simulation,
potentially only spatially spreading some of them (Mazzolini et al., 2024). This may
include the ingestion of under canopy in situ observations jointly with remotely
sensed retrievals of any kind. It is worth noting that, due to the assimilation of only
a single incomplete snow distribution map, the posterior simulations exhibit
equifinality (Beven & Freer, 2001), which prevents us from exploring in detail which
of these components is more dominant over the other since they are correlated
(Fig. 6). Adding other data sources and using more varied information could help
address this issue in future studies. In any case, the mean posterior values
obtained were close to unity for precipitation (in the physical space) and close to
zero for temperature, suggesting that it is not the total amount of precipitation
that is biased, but rather the small-scale redistribution of the meteorological
forcing.





Among the experiments that improved the simulations compared with the deterministic reference run, there was not a clearly superior experiment depending on the GC correlation length scale hyperparameter. Similar conclusions could be drawn from the findings in Cho et al. (2023), who tested different correlation length scales for their Gaussian decay-based localization function, showing that the differences were always lower than the improvement compared with their reference simulation. This suggests some flexibility in the choice of this hyperparameter, which may be complex especially when using non-Euclidean distances, and often limited by the availability of considerable computing resources. When comparing the Eu and Ma experiments, it was also difficult to spot differences if considering only quantitative error metrics. However, the spatial patterns at smaller scales seem more realistic when using the Ma configuration, as also found in Alonso-González et al. (2023). This is based on the fact that the snow spatial patterns are correlated with the characteristics of the terrain, since it controls its distribution by modulating accumulation and melt processes in both open and forested terrain (Geissler et al., 2024; Revuelto et al., 2014). It should be noted that the proposed domain is relatively small exhibiting a limited topographical complexity. Other experiments over larger areas of increasing topographical complexity may benefit from the increasing topographical variability. A potential limitation of this method will be found in non-complex terrain, as is typical in high latitude areas, where the topographical control of the snowpack dynamics may be less clear, although still very relevant (Bennett et al., 2022). In any case, snowpack in these areas exhibits less spatial variability, so we hypothesize that the use of Euclidean distance to map cell similarity is likely to be sufficient in these environments and/or at coarser resolutions. Alternatively, it is possible to use snow climatologies or observations to perform a more direct cell similarity mapping based on the persistence of the spatial patterns of the snow (Alonso-González, et al., 2023; Mazzolini et al., 2024). Despite the fact that developing snow cover climatologies in forest environments is significantly more challenging than in open terrain due to the aforementioned limitations of satellites to retrieve information beneath the canopy, it is possible to generate maps of the snow distribution in forested terrain by combining different techniques such as ground observations, lidar and field campaigns (Geissler et al., 2023). The generation of such products requires a significant effort in logistics that prevent its operational exploitation as a real time monitoring tool. In addition, such field methods will not be able to retrieve information at other times that the observation time itself. A promising application of the assimilation scheme presented here is to exploit such products to map the similarity between cells in forested terrain, allowing the significant effort needed for these initiatives to be



exploited to generate gap-free re-analyses or near real time updated simulations.
In this work, we have explored the effect of using the GC function to create a prior
covariance matrix in different spaces. However, what remains to be investigated is
the potential benefit of using different covariance (or kernel) functions. It is
possible that other functions may offer a more accurate representation of
snowpack correlograms across various spatial scales and resolutions, especially in
topographical Mahalanobis spaces. One obvious source of inspiration is to take
advantage of the extensive literature on kernels developed by the Gaussian
process community (Rasmussen & Williams, 2005). In particular, kernels with
compact support—those that become zero beyond a certain boundary— (Barber,
2020) could be of special interest since they will behave similarly to the GC
function, helping in limiting the computational cost and preventing spurious
correlations among the ensembles. Given the increasing availability of snow depth
information over large domains (Magnusson et al., 2024; Painter et al., 2016) , it
will be beneficial for the snow DA community to explore which kernel functions
better approximate the empirical snowpack spatial variability in different spaces
and resolutions. Given that snowpack exhibits persistent spatial patterns in both
forest and open terrain (Geissler et al., 2024; Helfricht et al., 2014), there is
potential to find a single flexible kernel configuration, ideally depending on a very
limited number of parameters, to be widely used in both spatiotemporal DA and
observation interpolation initiatives.

## 768 **5 Conclusions**

In this work, we have explored the potential of the observations obtained in forest
clearings to be used to update spatially complete snow simulations in forest
environments by means of spatio-temporal ensemble-based data assimilation. Six
different experiments were conducted in the Sagehen Creek (California, USA) using
different data assimilation configurations, demonstrating the potential obvious
benefits of spatiotemporal DA in forest environments. While most of the
experiments greatly improved the reference snow simulations, those relying on a
set of synthetic dimensions generated by a PCA were clearly inferior. Future
research may benefit from exploring other dimension reduction techniques such
as multidimensional scaling.
Among the remaining successful experiments, there was not a clearly superior
configuration , in that the differences among them were significantly lower than
the improvement compared with the reference run. This suggests some flexibility
on the selection of the critical hyperparameters of the DA. However, we found that



in terms of both qualitative and quantitative error metrics, those experiments built
on a cell similarity mapping based on the Euclidean distance were slightly more
accurate in terms of absolute validation metrics, but with a more realistic
representation of the spatial variance when using the Mahalanobis distance in a
topographical space. This suggests that this latter technique is better suited for
preserving spatial relationships in complex terrain. The critical differences found in
the implementation of a prior covariance function in different spaces, suggests the
importance of future research investing effort in development of specific kernels
with the aim of improving distributed snowpack simulations from spatially
incomplete observations in forested and/or complex terrain.
**Acknowledgments**
We acknowledge the Centre National d'Études Spatiales (French Space Agency,
CNES), for providing access to the supercomputing resources through the TREX
cluster. Esteban Alonso-González acknowledges funding from an European Space
Agency Climate Change Initiative (ESA-CCI) Research Fellowship (SnowHotspots
project). Kristoffer Aalstad acknowledges funding from the ERC-2022-ADG under
grant agreement No 01096057 GLACMASS, and an ESA-CCI Research Fellowship
(PATCHES project). Adrian Harpold and Cara Piske were supported by NSF EAR
#2012310 and EAR #1723990 to process and develop the datasets"
**Open Research**
MuSA (v2.2) is open source and can be found at Alonso-González et al. (2024). Future
versions of MuSA will be submitted to https://github.com/ealonsogzl/MuSA. The
assimilated airborne lidar snow depth data can be found at Piske (2022).
**Author Contribution**
Conceptualization was by EAG, AH, SG and JL. Methodology was by EAG and KA. Software
was by EAG, KA and LS. Validation and formal analysis was by EAG. Investigation was by
EAG and KA. Resources were provided by AH. Data Curation and visualization was by EAG.
Writing the original draft was led by EAG with key contributions from all authors. All
authors contributed to the review & editing of the original draft.

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
