# Peer review of "Ensemble-based data assimilation improves hyperresolution snowpack simulations in forests."

_EGUsphere, 2025_

## Referee Comment (RC2)

**Ensemble-based data assimilation improves hyperresolution snowpack simulations in forests.**

This exploratory manuscript addresses an important problem in snow hydrology, using ensemble-based data assimilation to improve snowpack simulations in forests. The work is technically rigorous; however, it would benefit from streamlining, particularly the Introduction, detailed methods, and the interpretation in Results/Discussion. My specific comments follow.

**General Comments:**

The introduction is comprehensive, but it is too long and repetitive, particularly in describing snowpack importance, canopy effects, observation challenges, and data assimilation (DA). Several paragraphs could be merged or trimmed without losing key context. Below are suggested line-numbered edits.

1. **Comment on Section 2.2:** This section is technically detailed and demonstrates careful implementation of the MuSA framework. However, it is very dense and at times reads more like a technical manual than part of a research article. For example, lines 272–322 (DES-MDA algorithm) provide a step-by-step explanation that is overly detailed for the main text; a shorter summary with reference to Alonso-González et al. (2022, 2023) would suffice. The description of MuSA v2.2 modifications (lines 392–426) is useful but reads like release notes, condensed to emphasize only the improvements directly relevant to this study or move it to supplement.

2. The authors should provide more detail on FSM2 parameterization under canopy (e.g., snow interception, sublimation, and radiation partitioning). Was the snow–canopy energy balance scheme modified from Essery et al. (2024)? Without this, it's difficult to judge whether the ensemble spread fully represents canopy–snow uncertainties.

3. **Comment on Sections 3 and 4:**
The results section clearly reports validation metrics, but it is heavy on numbers with limited interpretation. The discussion of Euclidean vs. Mahalanobis distance could be strengthened by explicitly linking improvements to spatial pattern representation. The evaluation also relies mainly on the posterior mean, which should be stated earlier, as it limits the interpretation of uncertainty-aware metrics (e.g., CRPS).

The discussion provides a thoughtful interpretation of the findings, but is dense and occasionally reads more like Methods. Details on computational cost and jitter regularization could be shortened or moved to a supplement. The PCA vs. Mahalanobis comparison is valuable but somewhat speculative; supporting diagnostics would add rigor. The key contribution that ensemble DA can propagate clearing observations under the canopy should be highlighted more prominently, with reduced repetition. Finally, the broader implications for operational snow monitoring and multi-source assimilation could be emphasized to connect more directly with the study's goal of improving under-canopy SWE estimates.

**Line to Line Comments:**

**Lines 54–83**: This section is informative but could be reduced by ~30–40% without losing clarity. The canopy–snow processes (lines 54–70) are somewhat repetitive; for instance, lines 60–66 (interception/unloading) and 67–70 (radiation effects) can be merged into a single concise description of the main mechanisms. Similarly, the discussion of monitoring challenges (lines 71–83) repeats the same idea across multiple sentences, lines 71–77 (direct observation challenges and harsh conditions) and lines 78–83 (lack of representativity, SWE vs. snow depth difficulty) could be combined into one tighter paragraph. Condensing these will improve readability.

**Lines 84–106:**
This section gives a thorough overview of remote sensing advances, but could be shortened by ~25–30% to avoid redundancy. Lines 84–90 (general remote sensing role and snow depth retrievals) and lines 91–94 (limitations in open terrain/temporal coverage) could be merged into one streamlined paragraph introducing both the capabilities and constraints of current methods. Similarly, lines 95–97 (SAR potential) and 98–99 (limitations in dense forests) can be combined into one sentence to avoid repeating "unfortunately." Finally, lines 100–106 (lidar under canopy) contain overlapping ideas: lines 100–103 (lidar partial penetration and validation) and 104–106 (refinements and future promise) could be merged into a single concluding paragraph on lidar as a partial solution.

**Lines 107–134:** This section is informative and relevant, but can use some massaging for improved readability. Lines 107–116 (model complexity and parameter uncertainty) and 117–125 (forcing and downscaling challenges) could be merged into a single concise discussion, while lines 126–134 (simplified downscaling and uncertainty propagation) could be shortened to avoid repeating the theme of uncertainty. I also recommend citing Raleigh et al. (2015) to emphasize how uncertainties from meteorological data propagate into snowpack simulations.

**135-136:** Good sentence!

137-138: Check for the font size and type.

**Lines 146–201:** This section is clear and well structured, but a few points could be tightened. The discussion of canopy-related observation challenges (lines 146–152 and 193–197) and DA limitations in forests (176–183 and 193–197) partly repeats the same idea and could be merged.

**2017-2018:** Details on how you improved the method would help readers.

**218-220:** Use correct citation format "Based on nearby SNOTEL at a similar elevation (SNOTEL Site: 539, Independence Camp, https://wcc.sc.egov.usda.gov/nwcc/site? sitenum=539, last accessed: 11-Nov-2024)"

**Figure 1:** Use (a),(b) … and improve the figure description for clarity.

**255:** Mention the section number instead of saying aforementioned

**270-271:** remove thanks to instead say something "due to model physics"

**275:** Mention the downscaled resolution of forcing (if they were forced after downscaling)

Figure 2: Adjust legend (place it outside of main figure for clarity).

---

## Author Response (AR1)

**REVIEWER 1**

**Review comments below are reproduced in blue and responses are in black.**

**General comments**

This study explores whether assimilation of remotely sensed snow depth observations available for forest clearings improve snowpack simulations below forest canopies where these measurements are missing. The authors performed six different data assimilation experiments using various configurations that affect prior correlations, and thereby the ability of the data assimilation schemes to propagate information from observed (open) to unobserved (forested areas) locations. The results show that four out of the six experiments improved the simulations in forested areas compared to the reference simulation. In the discussion, the authors provide an informative judgment of the results and specify future research possibilities for improving their methods further. Overall, the research presented in this study is highly relevant, as snow in forests can be important for a large range of scientific (e.g., ecological studies) and practical (e.g., water resources management) applications. The methods presented here are at the forefront of snow data assimilation research and demonstrate promising results. The study is well-written and provides valuable insights, and is therefore an excellent study that only requires a few minor adjustments before eventual publication in my opinion.

We sincerely thank the reviewer for their positive and encouraging feedback on our work. We are glad that the study's relevance and methodological contributions were clearly conveyed, and we appreciate the recognition of its potential applications.

We believe these improvements will further strengthen the clarity and impact of the manuscript. Below we provide a point-by-point response to each of the comments.

**Specific comments**

L 71-83: I think this paragraph can be shortened and should focus on why snowpack monitoring in forests, in particular below forest canopies, is challenging.

We will remove most of this paragraph and merge the relevant parts with the following one, which discusses the challenges of remote sensing in forests.

L 84-134: I recommend to shorten these two paragraphs too since the introduction is rather verbose. Just state the main problems concisely with references to the extensive literature, such as on challenges with forcing data as one example.

These two paragraphs will be simplified, removing details that are not relevant to the work.

L 42-201: Overall, I find the introduction a bit long and verbose. Please shorten where possible.

Several paragraphs will be shortened, simplifying the text where possible without changing the main message.

L 209-214: I don't understand this sentence. Please clarify.

We will simplify this as follows: "From all the available areas, we have manually selected a domain of ~2x2 km that maximizes canopy heterogeneity and the observed snowpack data that are incomplete due to dense canopy cover"

L 251-253: Please provide a scientific valid justification why this model was selected. Technical simplicity is not enough in my opinion. Why is the model appropriate for the experiments performed in this study? Which previous studies supports the choice of this model for the particular region, snow conditions and canopy properties?

The methods presented here are independent of the model and forcing used (as stated in Discussion). The 'better' the model and forcing (and observations), the easier the data assimilation will be. Each application will require the use of a different model. In any case, FSM2 is a widely known model, with a multitude of scientific and operational applications as outlined in Essery et al. (2025), which maintains a very good compromise between accuracy and computational cost. We will clarify this with the following sentence:

"In any case, the methods presented here are independent of the numerical model and forcing used, so they are transferable to different snow data assimilation initiatives."

L 504-506 and Table 1: I assume this table is for the snow depths of the canopy-covered cells? This is what L 231-233 states. Nevertheless, please specify this in the table caption and the text to avoid misunderstanding, or start the result section with repeating the information on L 231-233.

The referee is correct, we will clarify this in the caption as follows:

"Table 1: Validation metrics comparing the under canopy (withheld) and forest gaps (assimilated) observations"

L 602-606: Please add some more text here to guide the reader what the figure shows. It seems that low precipitation multipliers are associated with negative temperature adjustments, for instance. Is there an elevation trend in the values for precipitation multipliers and temperature additions?

This figure should be interpreted with caution. We are assimilating a single spatially incomplete observation, the amount of information available is limited. This correlation between precipitation and temperature is probably subject to equifinality. It is possible, and very interesting, to use data assimilation to identify weaknesses in numerical models by identifying irrelevant, highly sensitive or uncertain parameters. However, this would require a very different experimental design, with much more informative observations. Furthermore, this must be analysed jointly with the uncertainty of each inferred parameter, not just its posterior mean as shown here. We believe that this exceeds the scope of this work, and we will add the following sentence to clarify this:

"Figure 6 should be interpreted with caution. It is designed to provide a rough estimate of the posterior parameter values. However, drawing conclusions beyond that is risky, since there is likely to be equifinality in the parameter posteriors of the simulations, something

that is merely suggested by the obvious correlation between the posterior mean parameters."

L 623-636: It would be interesting to know how much time was spent running the model and how much time was needed for the DA algorithm separately. If I understand correctly, you run 100 ensembles with 40401 cells, and repeat this simulation 4 times in the iterative framework. Correct?

In this experiment, we generated ensembles consisting of 100 members (FSM2 realisations) for the 40,401 cells and repeated the process in four iterations. In any ensemble data assimilation algorithm, the greatest computational cost is due to the numerical model itself (or associated operations such as I/O). The analysis itself has a negligible cost, except in extreme cases where thousands of observations are assimilated (here we only assimilate a few for each cell) and therefore the cost of the domain localized linear algebra operations may be no longer negligible.

L 669-747: Maybe split these two long paragraphs into shorter ones.

We will separate these two paragraphs into simpler paragraphs.

L 788-792: Please simplify this sentence since it is hard to read and understand.

We will simplify as follows:

"The differences found in the implementation of the prior covariance function in the Mahalanobis and Euclidean spaces, suggests the importance of future research investing effort in exploring of specific covariance function that better capture the snowpack spatial patterns"

**Technical comments**

L 137: Capital letter after comma.

L 186-188: "Available" twice. Remove one.

L 209: "snow-off"?

L 275: What is "2m air fields"?

L 288: Missing period.

L 308: I guess "using" is missing.

L 364: On period too much.

L 475: Abbreviation PCA has not been introduced.

Thanks for spotting these typos, they will all be corrected.

**REVIEWER 2**

**Review comments below are reproduced in blue and responses are in black.**

This exploratory manuscript addresses an important problem in snow hydrology, using ensemble-based data assimilation to improve snowpack simulations in forests. The work is technically rigorous; however, it would benefit from streamlining, particularly the Introduction, detailed methods, and the interpretation in Results/Discussion. My specific comments follow.

We appreciate the reviewer's positive comments and constructive feedback. Below is a point-by-point response to each of the suggestions.

**General Comments:**

The introduction is comprehensive, but it is too long and repetitive, particularly in describing snowpack importance, canopy effects, observation challenges, and data assimilation (DA). Several paragraphs could be merged or trimmed without losing key context. Below are suggested line-numbered edits.

We will reduce the length of the introduction, also in line with Reviewer 1's suggestions.

**1. Comment on Section 2.2:**
This section is technically detailed and demonstrates careful
implementation of the MuSA framework. However, it is very dense and at times reads more like a technical manual than part of a research article. For example, lines 272–322 (DES-MDA algorithm) provide a step-by-step explanation that is overly detailed for the main text; a shorter summary with reference to Alonso-González et al. (2022, 2023) would suffice. The description of MuSA v2.2 modifications (lines 392–426) is useful but reads like release notes, condensed to emphasize only the improvements directly relevant to this study or move it to supplement.

We prefer to keep the description of the algorithm as it is included based on previous feedback from readers interested in more details on the implementation of the snow data assimilation algorithm. What we have included in the text is a simplified version, to give a general idea of how the assimilation scheme works. On the one hand, it is true that we have already detailed the algorithm in previous publications with great rigour, so curious readers can find this information in the relevant publications. However, we would like to avoid readers who want a general idea of how it works having to refer to other publications, which are significantly more dense than the few equations used here. Readers who are not interested at all can simply skip this relatively short part.
Similarly, we find it interesting to maintain the description of the modifications of MuSA in lines 392-426. These implementation details are the result of our experience and extensive performance testing, topics that are rarely discussed in data assimilation literature (even if they are critical). Specifically, the introduction of sparse linear algebra algorithms, k-d tree space-partitioning data structures, and how these elements interact with Mahalanobis space may serve as inspiration to other authors to develop tractable assimilation strategies based on our experience. Also, the description is necessary to ensure the reproducibility of the

results by others. We will still try to shorten this section where this is possible, while keeping the main ideas.

**2.** The authors should provide more detail on FSM2 parameterization under canopy (e.g., snow interception, sublimation, and radiation partitioning). Was the snow–canopy energy balance scheme modified from Essery et al. (2024)? Without this, it's difficult to judge whether the ensemble spread fully represents canopy–snow uncertainties.

We will include a description of the FSM2 parameterisation. It is important to highlight that the methods described here are independent of the model and forcing used. The more realistic the model and forcing, the easier the implementation from the snow data assimilation point of view. This is now explained in response to a similar comment from Reviewer 1. Regarding the FSM2 parametrizations, we will include the following sentence:

"The most complex FSM2 parameterisation was selected, based on previous experience. In the case of canopy parameterisation, this includes a two layer canopy model with nonlinear snow interception, snow unloading dependent on wind or temperature and two-stream approximation canopy radiative transfer."

**3. Comment on Sections 3 and 4:**
The results section clearly reports validation metrics, but it is heavy on numbers with limited interpretation. The discussion of Euclidean vs. Mahalanobis distance could be strengthened by explicitly linking improvements to spatial pattern representation. The evaluation also relies mainly on the posterior mean, which should be stated earlier, as it limits the interpretation of uncertainty-aware metrics (e.g., CRPS).

The results section includes all the validation metrics, which are discussed in the Discussion section. The alternative would be to merge the two sections into Results and Discussion, but we prefer the current scheme, which we consider more appropriate. All metrics were considered in the validation. Uncertainty is discussed when possible (e.g. CRPS or Figure 5). However, it should be understood that some metrics (and figures) can only be calculated (plotted) using optimal point-estimates, which is why we use the posterior mean which can be motivated from a decision theoretic point of view as the minimum mean squared error estimate (see Murphy, 2023). This is a necessary concession and very common in data assimilation and broader Bayesian literature.

The discussion provides a thoughtful interpretation of the findings, but is dense and occasionally reads more like Methods. Details on computational cost and jitter regularization could be shortened or moved to a supplement. The PCA vs. Mahalanobis comparison is valuable but somewhat speculative; supporting diagnostics would add rigor. The key contribution that ensemble DA can propagate clearing observations under the canopy should be highlighted more prominently, with reduced repetition. Finally, the broader implications for operational snow monitoring and multi-source assimilation could be emphasized to connect more directly with the study's goal of improving under-canopy SWE estimates.

At its core, computational cost is the reason for developing data assimilation algorithms. Without this enormous limitation, other non-tractable MCMC algorithms (or even brute force, such as grid approximations) would be the best option in terms of inferring parameters. We

believe that a paragraph describing the current computational limitations and giving advice on how to reduce this cost would be appreciated by readers considering developing data assimilation initiatives. There is only one sentence about jitter regularisation in the entire discussion, which we consider adequate. Unfortunately, it is not easy to find the reason why PCA has not performed as expected. However, this finding is in line with previous discoveries (as mentioned in the discussion*), which suggests that PCA distorts spatial relationships.

*"We hypothesize that the limitations found may come from the fact that the new set of coordinates do not explicitly preserve the Cartesian northing and easting information by mixing them with other dimensions, relaxing the relations between nearby cells in the Euclidean space (Davis & Curriero, 2019)"

We believe it is valuable to comment on this, even with limitations in the underlying reasoning, to warn future data assimilation scientist/engineers about the limited success of this technique, despite its obvious benefits from the implementation point of view.

The key contribution of the work, demonstrating that ensemble based DA improves snowpack simulations in forests, it is the first sentence of the discussion section "*The results shown here demonstrate the potential of ensemble-based DA experiments to improve hyper resolution snowpack simulations in forested terrain, by updating the canopy covered cells from information retrieved in clearings.*" Similarly, multisource assimilation (joint assimilation) is discussed around line 694, and in our experience working on developing this method we consider that the necessary "computational cost discussion" covers the main constraints for the operational implementation of these techniques.

**Line to Line Comments:**

**Lines 54–83:** This section is informative but could be reduced by ~30–40% without losing clarity. The canopy–snow processes (lines 54–70) are somewhat repetitive; for instance, lines 60–66 (interception/unloading) and 67–70 (radiation effects) can be merged into a single concise description of the main mechanisms. Similarly, the discussion of monitoring challenges (lines 71–83) repeats the same idea across multiple sentences, lines 71–77 (direct observation challenges and harsh conditions) and lines 78–83 (lack of representativity, SWE vs. snow depth difficulty) could be combined into one tighter paragraph. Condensing these will improve readability.

Several paragraphs from the introduction will be shortened following the recommendations.

**Lines 84–106:**
This section gives a thorough overview of remote sensing advances, but could be shortened by ~25–30% to avoid redundancy. Lines 84–90 (general remote sensing role and snow depth retrievals) and lines 91–94 (limitations in open terrain/temporal coverage) could be merged into one streamlined paragraph introducing both the capabilities and constraints of current methods. Similarly, lines 95–97 (SAR potential) and 98–99 (limitations in dense forests) can be combined into one sentence to avoid repeating "unfortunately." Finally, lines 100–106 (lidar under canopy) contain overlapping ideas: lines 100–103 (lidar partial

penetration and validation) and 104–106 (refinements and future promise) could be merged into a single concluding paragraph on lidar as a partial solution.

Similarly, the remote sensing section will be shortened according to referee recommendations, keeping the relevant parts.

**Lines 107–134:** This section is informative and relevant, but can use some massaging for improved readability. Lines 107–116 (model complexity and parameter uncertainty) and 117–125 (forcing and downscaling challenges) could be merged into a single concise discussion, while lines 126–134 (simplified downscaling and uncertainty propagation) could be shortened to avoid repeating the theme of uncertainty. I also recommend citing Raleigh et al. (2015) to emphasize how uncertainties from meteorological data propagate into snowpack simulations.

We will shorten this section and include the Raleigh et al. (2015) citation.

**135-136:** Good sentence!

We thank the referee for this positive comment

**137-138:** Check for the font size and type.

Corrected.

**Lines 146–201:** This section is clear and well structured, but a few points could be tightened. The  discussion of canopy-related observation challenges (lines 146–152 and 193–197) and DA limitations in forests (176–183 and 193–197) partly repeats the same idea and could be merged.

We are not entirely sure what the reviewer is referring to:
- Lines 146-152 discuss observation challenges in forests
- Lines 176-183 describe the limitations of previous works
- Lines 193-197 are the objectives of the work.

Since the topics are different, this is difficult to merge.

**217-218:** Details on how you improved the method would help readers.

We will add the following explanation: "We use a slightly improved method to extract vegetation from the snow surface described in  Piske (2025) that better resolves low vegetation from the snow surface using the lidar point cloud. "

**218-220:** Use correct citation format "Based on nearby SNOTEL at a similar elevation (SNOTEL Site: 539, Independence Camp, https://wcc.sc.egov.usda.gov/nwcc/site? sitenum=539, last accessed: 11-Nov-2024)"

We believe this is the correct format citation for a website

Figure 1: Use (a),(b) … and improve the figure description for clarity.

We will include identification letters to the subplots and caption

255: Mention the section number instead of saying aforementioned

We will remove aforementioned

270-271: remove thanks to instead say something "due to model physics"

Will use "due to model physics"

275: Mention the downscaled resolution of forcing (if they were forced after downscaling)

The resolution is 10m, this is explained in 2.1 Observed snow depth maps, vegetation parameters and meteorological forcing

Figure 2: Adjust legend (place it outside of main figure for clarity).

We will adjust the legend